Significant improvement in survival outcomes of trisomy 18 with neonatal intensive care compared to non-intensive care: a single-center study

Koshida Shigeki koshida@belle.shiga-med.ac.jp
Takahashi Kentaro
Perinatal Center, Shiga University of Medical Science , Tsukinowa-cho, Seta, Otsu, Shiga , Japan
Fujioka Kazumichi
Electronic publication date: 2023 Nov 29
Publication date: 2023
Volume: 11
Electronic Location ID: e16537
Received 2023 Aug 30; Accepted 2023 Nov 7
Copyright: © 2023 Koshida and Takahashi
Copyright year: 2023
Copyright holder: Koshida and Takahashi
License: This is an open access article distributed under the terms of the Creative Commons Attribution License, which permits unrestricted use, distribution, reproduction and adaptation in any medium and for any purpose provided that it is properly attributed. For attribution, the original author(s), title, publication source (PeerJ) and either DOI or URL of the article must be cited.
License URL: https://creativecommons.org/licenses/by/4.0/

Keywords: Trisomy 18, Neonatal intensive care, Survival outcomes, Palliative care, Parental autonomy

Funding: JSPS KAKENHI JP21K10494 This work was supported by JSPS KAKENHI Grant Number JP21K10494. The funders had no role in study design, data collection and analysis, decision to publish, or preparation of the manuscript.

==============================
Background

Trisomy 18 syndrome, also known as Edwards syndrome, is a chromosomal trisomy. The syndrome has historically been considered lethal owing to its poor prognosis, and palliative care was primarily indicated for trisomy 18 neonates. Although there have been several reports on the improvement of survival outcomes in infants with trisomy 18 syndrome through neonatal intensive care, few studies have compared the impact of neonatal intensive care on survival outcomes with that of non-intensive care. Therefore, we compared the survival-related outcomes of neonates with trisomy 18 between intensive and non-intensive care.

Methods

Seventeen infants of trisomy 18 admitted to our center between 2007 and 2019 were retrospectively studied. We divided the patients into a non-intensive group (n = 5) and an intensive group (n = 12) and evaluated their perinatal background and survival-related outcomes of the two groups.

Results

The 1- and 3-year survival rates were both 33% in the intensive group, which was significantly higher than that in the non-intensive group (p < 0.001). Half of the infants in the intensive care group were discharged alive, whereas in the non-intensive care group, all died during hospitalization (p = 0.049).

Conclusions

Neonatal intensive care for neonates with 18 trisomy significantly improved not only survival rates but also survival-discharge rates. Our findings would be helpful in providing 18 trisomy neonates with standard neonatal intensive care when discussing medical care with their parents.

Introduction

Trisomy 18 (T18) syndrome, also known as Edwards syndrome, is the second most common autosomal trisomy, after trisomy 21/Down syndrome (Cereda & Carey, 2012; Edwards et al., 1960). The syndrome is associated with an increased risk of poor perinatal outcomes including fetal death, stillbirth, and neonatal death, and as well as significant psychomotor and cognitive disabilities after infancy (Cereda & Carey, 2012). Although T18 has historically been considered lethal owing to its poor prognosis, with palliative care had been primarily indicated for T18 neonates (Bos et al., 1992; McGraw & Perlman, 2008), several recent reports have shown that T18 neonates receiving neonatal intensive care survived longer than those in previous studies (Kaneko et al., 2008, 2009; Kosho et al., 2006; Maeda et al., 2011; Nishi et al., 2014).

There have been many reports on the survival outcomes of T18 patients, with results varying depending on study design. The 1-year survival rate for T18 neonates was 8.7–12.6% in population-based studies (Nelson et al., 2016; Rasmussen et al., 2003; Suto, Isayama & Morisaki, 2021), whereas it increased to 25–59% in center-based studies implementing neonatal intensive care as the primary medical policy for T18 neonates (Kato et al., 2019; Kosho et al., 2006; Tamaki et al., 2022). Although neonatal intensive care for T18 neonates potentially improves survival rates, few studies have compared the impact of neonatal intensive care and non-intensive care on survival outcomes. When healthcare providers discuss the medical management plan with parents based on the natural history of their T18 neonate, it is essential to show the survival outcomes of T18 neonates receiving intensive care compared to those without intensive care.

Therefore, we compared the survival-related outcomes of T18 neonates, including survival and discharge rates, between those receiving neonatal intensive care and non-intensive care after birth.

Materials and Methods

Study design and ethical statement

This was a single-center retrospective observational study. This study was approved by the Institutional Review Board of Shiga University of Medical Science on December 1, 2021 (Approval No. R2021–144).

Data collection

We enrolled patients with karyotypically confirmed T18 before or after birth who were admitted to our neonatal intensive care unit (NICU) within the first month of life between 2007 and 2019. We excluded one patient with partial trisomy. Our medical care policy for T18 based on the best interest of the infant was determined according to the shared decision-making method, which respects parental autonomy. When an infant or fetus is diagnosed with T18, we offer three medical care policies for the infant: palliative, restrictive, and intensive care. We defined restrictive care as non-intensive care excluding highly invasive management such as full resuscitation at birth, respiratory management under endotracheal intubation, and surgical procedures. In this study, we classified infants receiving palliative or restrictive care as the non-intensive group and those receiving intensive care as the intensive care group.

Statistical analyses

Continuous variables are shown as medians (interquartile ranges) and were assessed using the Mann-Whitney U test. Categorical data are shown as n (%), and differences were assessed using the chi-square test. Survival rates for 36 months were analyzed using Kaplan-Meier curves and comparisons between the two groups were performed using a log-rank test. Statistical significance was set at P < 0.05. All statistical analyses were performed using IBM SPSS software program (version 22.0; IBM Japan, Tokyo, Japan).

Results

Perinatal background, major anomaly, surgical treatment, and survival discharge

Excluding one infant with partial T18, a total of 17 infants with full T18 were analyzed, including five classified into the non-intensive group and 12 classified into the intensive group (Fig. 1). Table 1 shows the perinatal background, major anomalies, surgical treatment, and survival to discharge of the T18 infants in this study. The percentage of Caesarean deliveries was significantly higher in the intensive group than in the non-intensive group. The Apgar score at 1 min was significantly lower in the non-intensive group than in the intensive group, whereas there was no marked difference in the 5 min Apgar score between the two groups. The other factors in the perinatal background showed no significant differences between the two groups. The proportion of infants who underwent surgical treatment was significantly higher in the intensive group than in the non-intensive group. All infants in the non-intensive care group died during hospitalization, whereas half of the infants in the intensive care group were discharged alive (p = 0.049).

Figure 1 Outline of this study.

T18 infants who received palliative or restrictive care were classified into the non-intensive group, while those who received intensive care were classified into the intensive group.

Table 1 Descriptive data of 18 trisomy infants, and comparison of perinatal background, anomaly, treatment and discharge according to medical care.

	All
(n = 17)	Non-intensive
(n = 5)	Intensive
(n = 12)	p	
Perinatal background					
G.A. at birth (weeks)	37.6 (37.3–40.2)	37.6 (37.5–37.6)	37.5 (36.5–40.4)	0.40	
Birth weight (g)	1,806 (1,644–1,917)	1,732 (1,722–1,844)	1,814 (1,606–1,936)	0.83	
Gender (male)	5 (29%)	1 (20%)	4 (33%)	0.08	
Mode of delivery (c/s)	8 (47%)	0 (0%)	8 (66%)	0.01	
Out-born	3 (18%)	1 (20%)	2 (17%)	0.87	
Prenatal diagnosis	8 (47%)	4 (80%)	4 (33%)	0.08	
Apgar score at 1 min	4 (3–5)	2 (1–4)	4 (3.8–5.5)	0.04	
Apgar score at 5 mins	7 (6–8)	5 (4–6)	4 (6–8.3)	0.06	
Major anomaly					
Cardiac diseases	17 (100%)	5 (100%)	12 (100%)	0.085	
Esophageal atresia	3 (18%)	1 (20%)	3 (25%)	0.82	
Surgical treatment					
Cardiac surgery	5 (29%)	0 (0%)	5 (42%)	0.09	
GI surgery	3 (18%)	0 (0%)	3 (25%)	0.22	
Any surgery	8 (47%)	0 (0%)	8 (67%)	0.01	
Survival discharge	6 (35%)	0 (0%)	6 (50%)	0.049	
Note:

GA, gestational age; c/s, Caesarean section; GI, gastrointestinal.

Survival rate

All five infants in the non-intensive-care group died within 1 month, while eight infants (67%) in the intensive care group died, and the remaining four (33%) survived for 3 years. The median survival time was 0 and 7 months in the non-intensive and intensive care groups, respectively. Figure 2 shows the Kaplan-Meier survival curves for each group. The 6-month survival rate was 66.7% (95% confidence interval CI [40.0–93.3%]), and the survival rates at 12 and 36 months were both 33%. (95% CI [6.7–60.0%]) in the intensive care group. The survival rate was significantly different between the two groups (p < 0.001).

Figure 2 The Kaplan-Meier survival curve of the 3-year survival of 18 trisomy infants.

The 6-month survival rate was 66.7% (95% CI [40.0–93.3%]), and the 36-month survival rate was 33.3% (95% CI [6.7–60.0%]) in the intensive group. The survival rate was significantly different between the two groups (p < 0.001).

Discussion

Reviewing T18 infants cared for in our NICU, we revealed that neonatal intensive care for T18 improved survival rates for three years and also found an increased rate of survival to discharge compared to non-intensive care.

First, we revealed that neonatal intensive care for T18 neonates improved their survival rates. The 1-year survival rates (33%) with neonatal intensive care in the present study is close to that of a previous study (25%) (Kosho et al., 2006) but much lower than those of the studies by Kaneko et al. (2009) and Tamaki et al. (2022) (47% and 59%, respectively). One possible explanation for the difference in survival rates between the present study and that of Kaneko et al. (2009) is the patient selection bias. Kaneko et al.’s (2009) study included T18 infants who underwent cardiac surgery, whereas we included all T18 infants who received intensive care regardless of cardiac surgery. A potential explanation for the difference in survival rates between the present study and that of Tamaki et al. (2022) is the indication for surgical treatment. Tamaki et al. (2022) indicated that the remarkable improvement in the survival of T18 patients in the late study period (2013–2017) could be attributed to the high surgical rates compared to those in the early period (2008–2012). The 1-year survival rate of 46% and 3-years survival rate of 25% for their total study periods (2008–2017) were closer to the present findings (33% for both 1- and 3-year survival rates). Our finding of a low survival rate with non-intensive care is consistent with Subramaniam et al.’s (2016) study, which indicated a median survival time of 0 days in T18 infants who received no interventions, as well as with Dereddy et al.’s (2017) study, which showed a 1-year survival rate of 3.9%. A significantly higher survival rate with intensive care for T18 neonates than with non-intensive care would justify providing standard neonatal intensive care to T18 neonates, as we do for critically ill neonates without T18 after birth. At the same time, it could be ethically unacceptable to not provide standard intensive care to a neonate simply because of the T18 diagnosis.

Next, we found that neonatal intensive care for trisomy 18 increased the rate of survival to discharge compared to non-intensive care. Consistent with Cortezzo, Tolusso & Swarr (2022), the survival to discharge rate for T18 infants with neonatal intensive care was significantly higher than that for those who did not receive intensive care. Consistent with recent studies (Kato et al., 2019; Tamaki et al., 2022), approximately half of the T18 infants in the neonatal intensive care unit were discharged alive. In addition to the survival period, the survival-discharge rate is also considered an important infant survival outcome that helps parents make decisions about their infant’s care.

Showing that intensive care for T18 infants improves the survival and survival-discharge rates compared to non-intensive care, our results suggest that T18 is no longer a uniformly lethal disease, as T18 infants are capable of being discharged home. Many perinatal professionals hold strong beliefs that the quality of life of individuals with severe disabilities is so poor that it would be unethical to prolong their lives (Pallotto & Lantos, 2017). Despite the severe developmental delay in T18 children, their parents had a positive attitude toward their care, and these children seemed to interact well with their parents and siblings (Kosho et al., 2013). Healthcare providers should convey recent data concerning the improvement of T18 survival outcomes through neonatal intensive care to parents whose fetuses or neonates have T18, thus helping parents make decisions regarding intensive care for these patients. Healthcare providers could present policies that make neonatal intensive care the standard of care for T18 neonates according to their respective institutional policies.

Interestingly, the rate of prenatal diagnosis of T18 was much higher in the non-intensive group than in the intensive group (80% vs. 30%). We therefore investigated the impact of a prenatal diagnosis on parental decision making regarding the medical care policy for their infant after birth. A prenatal diagnosis result of T18 could lead parents to be more likely to opt for non-intensive care rather than intensive care for their infant, considering the clinical data indicating a poor prognosis for T18 infants. This speculation could be supported by the official report on non-invasive prenatal tests in Japan, which indicates that 60.6% of pregnancies were terminated following a prenatal diagnosis of T18 (Steering Committee for Prenatal Testing Approval System in Japan, 2023). As we evaluated patients whose fetus was prenatally diagnosed to have T18 after 22 weeks of gestation, a stage in which termination is not permitted, it is understandable that most parents would select non-intensive care for their infant after birth instead of termination of their pregnancy.

Several limitations of the current study should be mentioned. First, the number of participants in this study was smaller than that in other recent studies (Cortezzo, Tolusso & Swarr, 2022; Kaneko et al., 2009; Kato et al., 2019; Tamaki et al., 2022) and thus may be less reliable for evaluating postnatal outcomes. Despite the small number of patients at a single center, we demonstrated an improvement in survival outcomes of T18 neonates with intensive care compared to non-intensive care within the same period, which was based on parental autonomy, whereas several studies have shown the improvement of survival outcomes by intensive care for T18 neonates compared to previous periods or with population-based data. Second, the study did not include fetal death patients of T18 with a prenatal diagnosis. Some of these fetal deaths may have been cases in which the parents might have requested intensive care for their neonates after birth. If they were included in the intensive group, the difference in postnatal outcomes from the non-intensive group might have changed.

Conclusions

We conclude that neonatal intensive care for T18 infants improves not only survival rates but also survival-discharge rates. Healthcare providers should be aware that neonatal intensive care for T18 is the standard of care and should discuss the medical management policy for T18 with parents before and after birth.

Supplemental Information

Supplemental Information 1 Clinical data used in the current study.

Click here for additional data file.

We thank the medical staff in the NICU of the Shiga University of Medical Science Hospital for their assistance with this study.

Additional Information and Declarations

Competing Interests

Author Contributions

Human Ethics

Data Availability

The authors declare that they have no competing interests.

Shigeki Koshida performed the experiments, analyzed the data, prepared figures and / or tables, and approved the final draft.

Kentaro Takahashi conceived and designed the experiments, authored or reviewed drafts of the article, and approved the final draft.

The following information was supplied relating to ethical approvals (i.e., approving body and any reference numbers):

This study was approved by the Institutional Review Board of Shiga University of Medical Science (Approval No. R2021–144).

The following information was supplied regarding data availability:

The clinical data used in the current study are shown as raw values in the Supplemental File.

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
