# Peer review of "Significant improvement in survival outcomes of trisomy 18 with neonatal intensive care compared to non-intensive care: a single-center study"

_PeerJ, doi:10.7717/peerj.16537_

## Round 0.1 · original submission · Major Revisions

One reviewer is preferable, and the other is not. Thus, please revise as carefully as you can to overcome all the arisen critiques.

·

Basic reporting

The paper is nicely composed. I have 2 suggested changes:
- in the 1st sentence of the abstract I would say, trisomy 18, also known as Edwards syndrome, is the second most common autosomal trisomy after trisomy 21/Down syndrome.
-in the Results avoid the term "case" when referring to infants and say patient, infant, or individual.

Experimental design

no comment

Validity of the findings

As authors point out the number of patients is small. However I think the comparison is valid, and they point out the issue of the N in the text.

Additional comments

I have 2 suggestions:
- others (Cortezzo et al., Goel et al.) have shown that prenatal diagnosis of trisomy 18 increases early mortality; I would point out in the Discussion that their non intensive group had 80% occurrence of prenatal diagnosis while the treated/ intervention group had 33%. Given the numbers of infants on the groups that difference may not be significant in a statistical comparison, but I would still do and discuss the issue. Discussion of why the prenatal diagnosis could increase the risk of mortality would be interesting as well.
-Cortezzo et al. also showed that the intervention group as opposed to the comfort care group had better survival although they don't discuss this difference. The reader can see the survival of the treated group in their figures.

Reviewer 2 ·

Basic reporting

Some grammatical mistakes, but overall OK. Made some strong statements that I believe are inaccurate that might be language related?
Missed some important literature references

Experimental design

Small retrospective review with expected outcomes based on known literature with much bigger numbers.
Didn't require anything very rigorous to review.

Validity of the findings

Not novel.
just replicated what is already know - if we try to prolong life we can.
Conclusions are quite broad and strong for this small of a sample.

Additional comments

Small sample
Demonstrates what other studies have already found.
Line 202 – “no longer a lethal disease” – how can we say this. Just as we maybe shouldn’t say it is always lethal, we can’t say that it isn’t lethal a majority of the time? This statement is just wrong.
Line 210 – I strongly disagree that providers should choose policies that make neonatal intensive care the standard. I think the standard should be that families get true information and make the decision whether they want to proceed or not.
Again, there are studies showing that most families who are given honest information about outcomes, both short and long term development, and not persuaded by parents who project a wonderful life, or those who project lethality, still choose termination or comfort care, and not major life prolonging interventions. Studies also demonstrate a survival of the fittest picture if we are honest with ourselves.
Line 217 – based on “parental autonomy” – this really depends on the counseling and whether truly honest versus persuasive one way or the other.

---

## Round 0.2 · accepted · Accept

I feel the revision was well done.

·

Basic reporting

The authors have aptly addressed my comments. The basic reporting is clear and concise.

Experimental design

The design is a retrospective cohort and fits the question. As mentioned by myself and the other reviewer the numbers are smaller than prior studies but add to the literature, and I would consider this replication important.

Validity of the findings

As mentioned the only limitation is the small number of patients.

Additional comments

I would recommend proceeding as more data are needed on this question.